# Electrostatic steering of thermal emission with active metasurface control of delocalized modes

Joel Siegel[1,5], Shinho Kim [2,5], Margaret Fortman[1], Chenghao Wan [3], Mikhail A. Kats [3], Philip W. C. Hon [4], Luke Sweatlock[4], Min Seok Jang [2] ✉ & Victor Watson Brar [1] ✉

We theoretically describe and experimentally demonstrate a graphene-integrated metasurface structure that enables electrically-tunable directional control of thermal emission. This device consists of a dielectric spacer that acts as a Fabry-Perot resonator supporting long-range delocalized modes bounded on one side by an electrostatically tunable metal-graphene metasurface. By varying the Fermi level of the graphene, the accumulated phase of the Fabry-Perot mode is shifted, which changes the direction of absorption and emission at a fixed frequency. We directly measure the frequency- and angle-dependent emissivity of the thermal emission from a fabricated device heated to 250 °C. Our results show that electrostatic control allows the thermal emission at 6.61 μm to be continuously steered over 16°, with a peak emissivity maintained above 0.9. We analyze the dynamic behavior of the thermal emission steerer theoretically using a Fano interference model, and use the model to design optimized thermal steerer structures.

The mid infrared (MIR) is an important band for applications ranging from free-space laser communications[1] to chemical sensing[2,3]. An optimal MIR source for these applications would be narrowband, and also offer high speed directional control, such that the beam can be rastered over a range of angles, or have a controllable focal point. Typically, such beam-steering is achieved by reflecting a beam using mechanical devices such as gimbal-mounted mirrors[4], optical phased arrays of antenna[5,6], or liquid crystal-based devices[7]. While each of these techniques have their own set of advantages and disadvantages, one limitation common to them all is that they require an external source of light, such as a quantum cascade laser.

An alternative source of MIR light is one that can be found everywhere, thermal radiation. Any material at a non-zero temperature will emit radiation over a broad range of frequencies which, at moderate temperatures (0–700 °C), is peaked in the MIR. Though thermal emission is typically viewed as incoherent,

isotropic, and broadband, recent advances in nanoengineering have demonstrated that it is possible to engineer the emissivity of a structured material to create narrowband[8] directional[9] emission that exhibits coherence. These include metasurfaces composed of non-interacting, localized resonator elements tuned to specific wavelengths, such as metallic nanoantennas[10] or semiconducting nanostructures that exhibit sharp quasi-bound state in the continuum resonances[11,12]. To achieve coherent directional emission, meanwhile, structures that support long-range delocalized modes can be utilized. These include surface waves that are out-coupled via gratings[9,13,14], Fabry-Perot (F-P) cavities[15], photonic crystals[16–18], epsilon near zero modes[19] and delocalized modes formed by coupled resonators[20–22]. In all of these demonstrated devices, heating is all that is required to produce the desired light as the relevant optical modes are excited thermally, thus providing an elegant source of MIR radiation.

[1]Department of Physics, University of Wisconsin-Madison, Madison, WI, USA. [2]School of Electrical Engineering, Korea Advanced Institute of Science and Technology, Daejeon, Republic of Korea. [3]Department of Electrical and Computer Engineering, University of Wisconsin-Madison, Madison, WI, USA. [4]Northrop Grumman Corporation, Redondo Beach, CA, USA. [5]These authors contributed equally: Joel Siegel, Shinho Kim. ✉e-mail: jang.minseok@kaist.ac.kr; vbrar@wisc.edu

Imparting tunability into such devices—which could allow for dynamic beam control and frequency shifting—requires the integration of materials with variable optical properties. Materials with temperature-dependent phases and/or indices, such as GST[23–25], VO2[26–29], or Si[5] have been utilized to create metasurfaces that control the magnitude and phase of scattered light in reflection or transmission geometries, but such materials are unsuitable for thermal emission devices that operate at high, constant temperatures. Alternatively, materials with indices that depend on carrier density, including graphene, III−V quantum wells and indium tin oxide (ITO), can be utilized to bestow electrostatic tunability on metasurfaces, and devices that control phase, frequency, and intensity of reflected light have been demonstrated[30–35]. These materials are also chemically and phase stable at high temperatures, which has enabled them to be integrated within thermal-control metasurfaces to electrostatically tune the intensity and frequency of incandescent light in the mid-IR[36–38]. Unfortunately, such materials also introduce ohmic loss which can, in some geometries, suppresses formation of the long-range delocalized modes that are necessary for coherent, directional thermal emission. As such, dynamic angular tuning of thermal emission is an outstanding problem in the field of thermal metasurfaces.

In this work, we theoretically describe and experimentally demonstrate a thermal emission device that can be tuned electrostatically to control the directionality of thermal emission within a narrow bandwidth. We show experimentally that by using a tunable graphene-integrated metasurface as a boundary for a delocalized F-P cavity mode, the thermal emission from a surface at 6.61 $\mu$m (1508 cm$^{-1}$) can be continuously steered by $\pm 16°$ by changing the carrier density of the graphene sheet. Theoretical calculations, meanwhile, show that an optimized geometry using real materials could achieve $\pm 60°$ of continuous tuning.

## Results

For dynamic thermal emission steering, we utilize an electrically tunable F-P resonance of a SiN$_x$ dielectric layer sandwiched by a gold back reflector and a graphene-based active metasurface as illustrated in Fig. 1a. The graphene metasurface consists of 30 nm thick, 1 $\mu$m wide gold strips spaced 40 nm apart on top of a HfO$_2$ (5 nm)/graphene/Al$_2$O$_3$ (30 nm) trilayer, sitting on the 2 $\mu$m thick SiN$_x$ membrane with the 100 nm gold back reflector that also serves as a back gate electrode. The gaps between the gold strips are filled with a bilayer of 30 nm gold and 100 nm SiO$_x$. The sub-wavelength period of the structure suppresses far-field diffraction except for the zeroth order. We note that this structure does support metal-insulator-metal (MIM) surface plasmon modes[39], but for the device dimensions used in this work the MIM resonances occur at frequencies (~4500 cm$^{-1}$) much higher than the active frequency (~1500 cm$^{-1}$), and thus have little effect on the thermal steering properties of the device.

The working principle of our device is illustrated in Fig. 1b. The graphene-based metasurface covering the top surface of the SiN$_x$ membrane acts as a partially reflecting mirror to form a vertical F-P cavity. By applying an electrostatic potential ($V_G$) across the dielectric spacer, the Fermi level of graphene ($E_F$) is modulated and so are the complex reflection and transmission coefficients of the top graphene metasurface. Consequently, the condition for the resonance shifts, causing a shift in the peak emission angle ($\theta$) for a given frequency. These changes can be qualitatively understood by treating the top metasurface as a two-dimensional sheet with an effective surface admittance, which is justified since the metasurface thickness is about

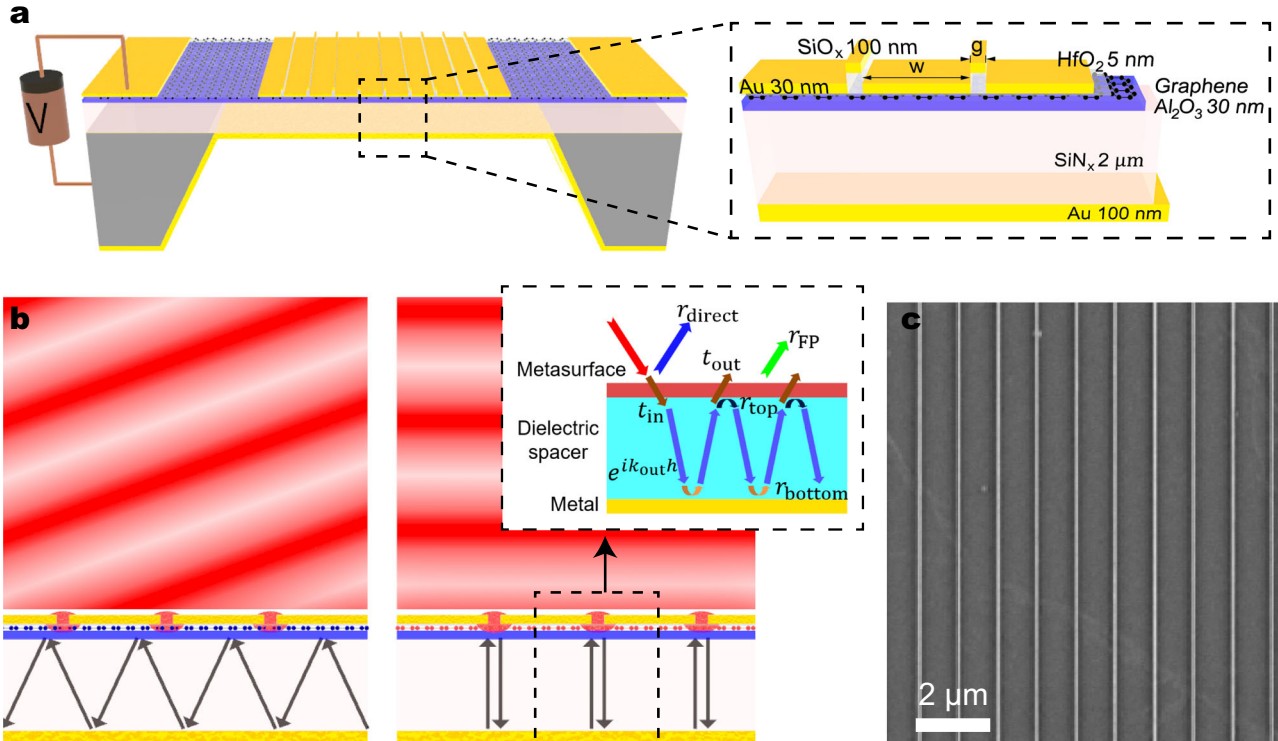

**Fig. 1 | Schematic and working principles of dynamic thermal emission steering device. a** Diagram of the thermal steering device. The magnified view shows the geometry of a graphene-Au slit metasurface unit cell. **b** Illustration of the working mechanism of electrically tunable directional thermal emission via graphene metasurface control of the delocalized Fabry-Perot modes in the dielectric. The emission angle of the structure is controlled by the incident-angle-dependent resonant absorption condition, which changes with the graphene Fermi level. The inset shows decomposed total reflection into two reflection channels: direct reflection $r_{direct}$ and F-P reflection $r_{FP}$. **c** Scanning electron microscopy image of the graphene metasurface on top of the SiN$_x$ membrane. The width (w) and gap (g) of the Au slit array are 1 $\mu$m and 40 nm, respectively.

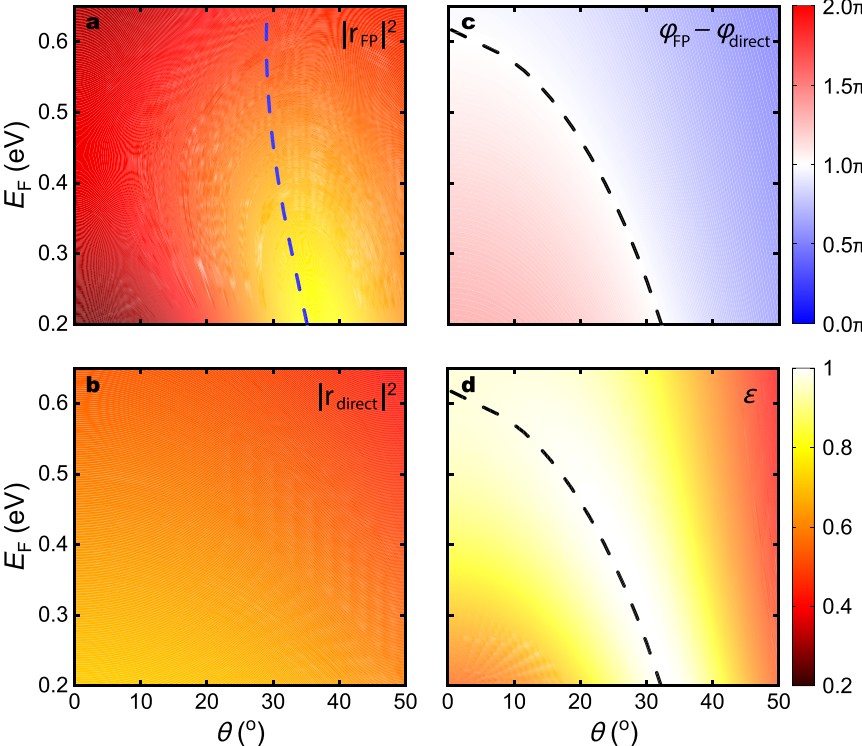

**Fig. 2 | Angular and Fermi-level dependence of reflection coefficients and emissivity.** Fermi-level and angular dependence of **a** the reflectance due to the F-B resonance ($|r_{FP}|^2$), **b** the direct reflectance from the top surface of the device ($|r_{direct}|^2$), **c** The phase difference between the two reflection ($\phi_{FP} - \phi_{direct}$), and **d** the emissivity of the device ($\epsilon$). The blue dashed line in **a** indicates the F-P resonance condition. The black dashed line in **c**, **d** indicates the condition for destructive interference between $r_{FP}$ and $r_{direct}$, $|\phi_{FP} - \phi_{direct}| = \pi$. All angular spectra are calculated for frequency $\omega = 1503$ cm$^{-1}$.

two orders of magnitude shorter than the wavelength of the free space light[6,30,40]. In this model, the subwavelength metallic stips with narrow gaps make the overall optical response of the graphene metasurface to be highly capacitive (i.e. large imaginary impedance) at a low carrier concentration. As the conductivity of graphene raises with increasing $E_F$, the metasurface exhibits a reduced, but still high, capacitance and also acquires a larger conductance, changing the reflection/transmission characteristics. The quantitative surface admittance model for the graphene metasurface is discussed in detail in Supplementary Notes 1, 2, and 3.

Recognizing the emissivity $\epsilon(\omega, \theta)$ of a reciprocal object is equal to its absorptivity $\alpha(\omega, \theta)$[41], one can understand the mechanism of the directional shift in thermal emission more intuitively by analyzing the absorption process. Since the transmission channel is blocked by the back reflector,

$$\epsilon = \alpha = 1 - |r_{tot}|^2 = 1 - |r_{direct} + r_{FP}|^2, \quad (1)$$

where $r_{tot}$ is the total reflection, which can be decomposed into the direct reflection from the top surface ($r_{direct}$) and the resonant reflection due to the F-P interference formed by multiple reflections inside the dielectric spacer ($r_{FP}$). The interplay between $r_{FP}$ and $r_{direct}$, both of which are dependent on $E_F$, determines the overall absorption (and thus the emission) of the device. The absorption peak occurs when $r_{FP}$ and $r_{direct}$ destructively interfere with each other by having similar amplitudes and a $\pi$ phase difference.

We first theoretically investigate the behavior of the proposed device using full-field electromagnetic simulations based on the finite element method as summarized in Fig. 2. The dependence of $r_{direct}$ on $\theta$ and $E_F$ for TM polarized light is shown in Fig. 2b. $r_{direct}$ can be obtained by simulating the reflection by the graphene metasurface sitting on a semi-infinite SiN$_x$ layer without a back reflector. Since the

top graphene metasurface does not support any distinctive resonance around the target frequency of $\omega = 1503$ cm$^{-1}$, the direct reflectance, $R_{direct} = |r_{direct}|^2$, exhibit a generic weak dependence on $\theta$ within the range of $0°$ to $50°$. As the carrier density of graphene increases, the metasurface becomes less capacitive, leading to better impedance matching as elaborated in (Supplementary Notes 1 and 3). Consequently, $R_{direct}$ monotonically decreases with increasing $E_F$. The phase of the direct reflection, $\phi_{direct} = \arg\{r_{direct}\}$, remains nearly constant round $0.9\pi$ within $\theta \in (0°, 50°)$ and $E_F \in (0.2, 0.65)$ eV.

Unlike $r_{direct}$, $r_{FP}$ shows a strong dependence on both $\theta$ and $E_F$ due to its resonant nature. The F-P resonance occurs when the out-of-plane wavevector inside the dielectric, $k_{out} = nk_0\sqrt{1 - \sin^2\theta}$, satisfies the constructive interference condition:

$$2k_{out}h + \phi_{top} + \phi_{bottom} = 2\pi m, \quad (2)$$

where $k_{out}h$ is the phase accumulation associated with vertical wave propagation across the dielectric spacer, $\phi_{top}$ and $\phi_{bottom}$ are the reflection phase from the top and bottom surfaces, respectively, and $m$ is an integer. $\phi_{bottom} \sim \pi$ does not depend on $E_F$ since the bottom surface is a mere gold back reflector, which behaves like a perfect electric conductor at mid-infrared frequencies. $\phi_{top}$, in principle, could depend on $E_F$ for metasurfaces with an admittance comparable to the surrounding medium, but in our device the admittance is large and, thus, the dependence of $\phi_{top}$ is weak for $E_F \in (0.2, 0.65)$ eV. (see Supplementary Note 1 for a detailed analysis). As a result, at a fixed frequency, the resonance angle $\theta_{FP}$ slightly decreases from $35°$ to $29°$ when $E_F$ increases from 0.2 eV to 0.65 eV as indicated as a blue dashed curve in Fig. 2a; And, at a fixed $\theta$, the resonance frequency $\omega_{FP}$ slightly blueshifts with increasing $E_F$. The F-P resonance becomes weaker with increasing $E_F$ as the top graphene metasurface becomes less reflective and more absorptive, raising both the radiative and dissipative decay

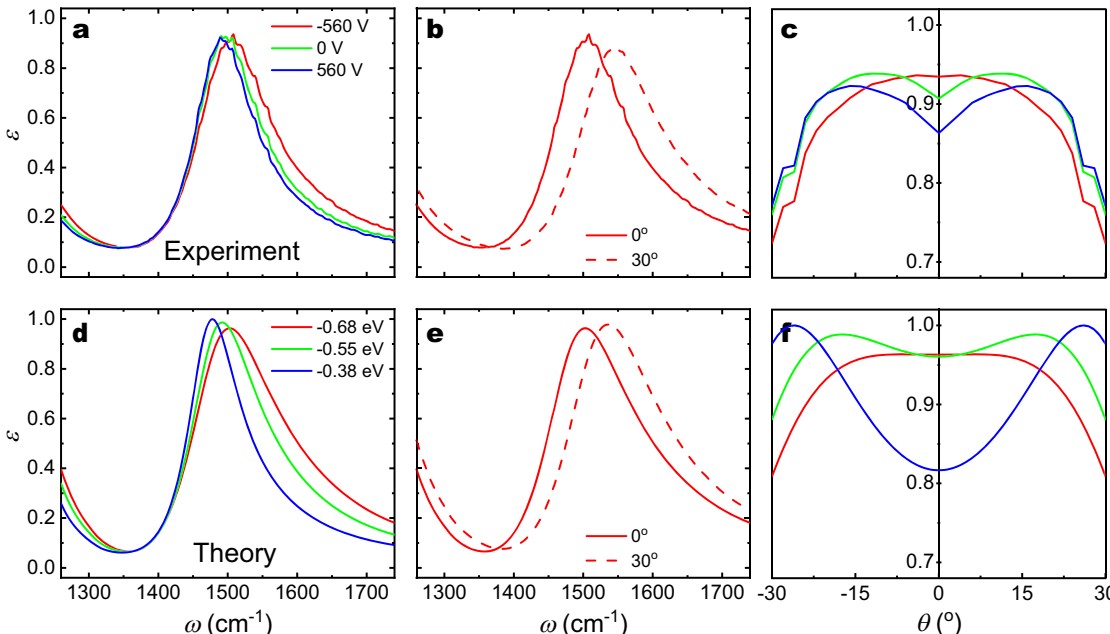

**Fig. 3 | Measured and calculated emissivity dependence on gate voltage and emission angle. a–c** Experimental emission spectra for **a** different applied gate voltages at normal incidence, **b** at different angles at a constant applied voltage of −560 V, and **c** the angular-dependent emission at 1508 cm⁻¹. Experimental measurements were obtained for 0° < θ < 30° and are mirrored for visual clarity. **d–f** Show analogous simulated spectra for equivalent values of $E_F$, with **e** plotted for $E_F = -0.68$ eV and **f** plotting angular-dependent emission at 1503 cm⁻¹.

rate of the resonant mode. However, while $\phi_{top}$ shows only a small dependence on $E_F$, the overall phase shift due to the F-P resonance ($\phi_{FP}$) includes phase accumulated while passing into and out of the F-P cavity, through the complex transmission coefficients $t_{in}$ and $t_{out}$, which show considerably more dependence on $E_F$. (see Supplementary Note 3).

Since the amplitude of $r_{FP}$ is similar to that of $r_{direct}$ near the broad F-P resonance, what mainly determines the overall absorption is their phase difference, $\phi_{FP} - \phi_{direct}$. We note that the Fano interference between a non-resonant and a resonant scattering channel has been widely adopted to create a sharp resonant response[42,43]. The dependence of $\phi_{FP} - \phi_{direct}$ on $E_F$ and $\theta$, which is dominated by $\phi_{FP}$ due to the near constant $\phi_{direct} \approx 0.9\pi$, are plotted in Fig. 2c. $\phi_{FP}$ monotonically decreases with $\theta$ because the propagation phase across the dielectric spacer, $k_{out}h$, decreases as $k_{out}$ shortens. $\phi_{FP}$ also decreases with $E_F$ as the capacitive phase shift of the top graphene metasurface reduces. As a result, the condition for the Fano resonance, $\phi_{FP} - \phi_{direct} = \pi$, shifts from $\theta_{res} = 32°$ to 0° as $E_F$ alters from 0.2 eV to 0.62 eV. This change in the phase matching condition drives an overall change in the angular-dependent absorptivity/emissivity, shown in Fig. 2d, and thus allows the device to thermally emit at an angle that can be tuned by varying $E_F$.

To experimentally verify the possibility of active thermal emission steering, we fabricated the proposed device using e-beam lithography over a 4 × 4 mm² area (see Methods), heated it to 250 °C, and measured its angle-dependent thermal emission spectra while varying the $E_F$ by applying different gate voltages $V_G$. A polarizer was used to accept only TM polarized emission, and the acceptance angle of the emitted light was 3°. The emissivity of the structure is calculated by normalizing the emitted radiation of the device to the emitted radiation of a reference carbon nanotube blackbody[44].

The measured surface normal emissivity spectra for $\theta = 0°$ at $V_G = 560$, 0 and −560 V, shown in Fig. 3a, exhibit a well-defined resonance peak at around 1500 cm⁻¹ that blueshifts as the Fermi level of graphene increases, indicating that the thermal emission peaks are electrostatically tunable with minor variation in the intensity. The measured emissivity spectra also shows a strong angular dependence as shown in

Fig. 3b. At a constant doping level ($V_G = -560$ V), the emission peak shifts from 1508 cm⁻¹ to 1543 cm⁻¹ as $\theta$ changes from 0° to 30°. There are also higher order features present around 2400 cm⁻¹ (see Supplementary Note 4) that show similar but more limited shifting. Finally, Fig. 3c demonstrates the dynamic thermal emission steering by showing how the emission angle is modulated by altering the doping level of graphene at a fixed target frequency $\omega = 1508$ cm⁻¹. At $V_G = -560$ V, we observe that the emission peak is most intense at normal incidence and decreases in intensity as the angle is increased. As the applied gate voltage increases to 560 V, the lobe shifts from normal incidence to increasing angles, up to 16°, allowing for continuous tuning in that range.

These experimental results can be compared to simulated emissivity spectra shown in Fig. 3d–f. In these simulations, the value of $E_F$ at $V_G = 0$ V was chosen as a fitting parameter and calculated spectra were compared to the experimental spectra obtained at $V_G = 0$ V to determine that $E_F = -0.55$ eV with no gate voltage applied. This indicates that the sample is heavily hole-doped, which is consistent with previous studies of graphene grown and transferred using similar procedures[45]. Using this initial value of $E_F$, the Fermi energies at other gate voltages were derived with a simple capacitance model.

The overall qualitative behavior of the simulations (Fig. 3d–f) is consistent with our experimental results (Fig. 3a–c), however, the emission lobes are broader and the change of emission angle of the emitter is smaller in our experiment than was theoretically predicted. The likely sources of these inconsistencies are the metastructure geometric and material parameter variations across the full 4 × 4 mm² device (see Subsections A and B in Supplementary Note 5), and carrier density variation during the heating process due to the temperature dependence of the SiN$_x$, Al$_2$O$_3$, and HfO$_2$ dielectric properties[46–51]. The estimated carrier density tuning range is also affected by substrate and interface charge traps, which can act to decrease the overall doping range by screening the applied gating field (see Methods); Moreover, the tuning range is directly affected by the DC dielectric constant of the SiN$_x$ layer, which has reported variations of 15% for the commercial membranes used in this work[52]. We also note that the modulation depth

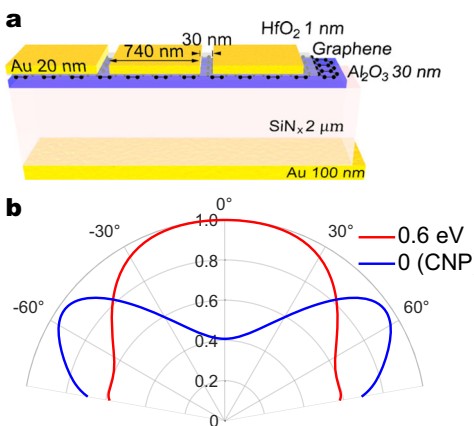

**Fig. 4 | Optimized steering of thermal emission. a** Schematic of the device geometry that maximizes thermal emission steering performance. **b** The angular emission spectra of the optimized device for Fermi levels of 0 and 0.6 eV at $\omega = 1614\ cm^{-1}$.

at $\theta \approx 0°$ is predicted to be larger than what is observed experimentally, and we also attribute this mostly to decreases in the doping range, as well as small potential misalignment of the heating stage (see Methods). The intensity of emission at large angles can also be reduced due to ellipsoidal elongation of the measurement area which, for small device areas, can extend the active zone to include some low emissivity, unpatterned gold areas. Finally, we note that the initial graphene doping level can affect the calculated spectra, but simulations show that such affects do not account for the discrepancies we observe (see Supplementary Note 5). Those simulation results do indicate, however, that the dynamic steering range could be improved in the future by using graphene with slightly less initial doping.

To further explore the potential performance of the proposed thermal steerer device, we investigate the maximum realizable emission angle under the limitation of realistic geometric and material parameters. The Fermi level of graphene is assumed to be electrostatically tunable between 0 eV and 0.6 eV, considering typical dielectric strength of SiN$_x$ and numerical optimizations of the geometric parameters of the device were performed to maximize the angle tunability. To prevent performance degradation due to non-local effects (see Supplementary Note 5 for more discussion), we set the minimum gap width to 30 nm and carried out simulations in the frame of classical electrodynamics. Figure 4a shows the structure of the optimized device. The gap and width of Au slit array are 30 nm and 740 nm, respectively. The HfO$_2$ is thinned to 1 nm which is achievable smallest value that could avoid quantum tunneling effect. The bilayer Au/SiO$_x$ area eliminated to enhance interaction between graphene and Au slit array. The optimization results show that it is possible to achieve ~ 60° thermal emission angle steering with unity peak emissivity (Fig. 4b). The achievable performance is greater than most metasurface-based electrically tunable beam steering devices[53] and is comparable to state-of-the-art MEMS-based beam steering devices with a 70° field of view[54]. The improvements in the optimized structure in comparison to the experimentally measured sample are due to three main effects. First, the optimized structure utilized a smaller, 30 nm spacing between the gold strips. This acts to increase the electric field concentration within the graphene and minimize stray fields connecting the gold strips, allowing more interaction with the graphene and a stronger effect of the graphene on the metasurface properties. Second, a thinner HfO$_2$ layer is used in the optimized structure, which brings the graphene closer to the gold and also increases the electric field intensity within the graphene sheet (see Supplementary Notes 2 and 5). And, third, in the optimized structure we assume a greater range of $E_F$ tunability, which is consistent with the potential

properties of the dielectrics, but could not be achieved in our experiments due to our methods of contacting the sample (i.e. wirebonding) which weakened the dielectric strength and restricted the range of $V_G$. We note that the required gate voltage for device operation can be significantly reduced by modifying the gating scheme. For example, as demonstrated in the work of N. H. Tu et al.[55] and B. Zeng et al.[56] by inserting a transparent conducting layer near the top graphene membrane, the gate voltage required to achieve the same level of Fermi energy can be reduced by orders of magnitudes with a marginal perturbation of the optical characteristics of the device.

## Discussion

In conclusion, we have demonstrated a thermal emitter that can continuously change the angle of emission in the mid-IR for a designated frequency. We show that by including a graphene-metal metasurface as a boundary, a delocalized F-P optical mode can be tuned to exhibit resonances with angular and frequency dependencies that depend on the carrier density of graphene, which can be tuned electrostatically. The net result is a surface that has an emissivity that is strongly angular dependent and tunable. 16° of thermal emission steering at 6.61 μm was demonstrated experimentally, and we outline design strategies that could increase the tunability to almost 60°. This work lays the foundation for next generation beamsteering devices that do not require an external light source, and could be broadly applicable for remote sensing and thermal camouflage applications.

## Methods

### Fabrication of Device

SiN$_x$ membranes (2 μm thick and 5 mm × 5 mm wide) on a 200 μm Si frame were purchased from Norcada. To the backside of the SiN$_x$ membrane, we deposited a 2.5 nm chromium adhesion layer followed by a 100 nm of gold, which makes the lower layer opaque and reflective. Atomic Layer Deposition (a Fiji G2 ALD) was used to grow a 30 nm film of Al$_2$O$_3$ on the top of the SiN$_x$ membrane. Once the Al$_2$O$_3$ was grown, a prepared graphene sheet was transferred on top of the Al$_2$O$_3$ film. Graphene was purchased from Grolltex and was grown on a Cu foil. To remove the foil, first a protective layer of PMMA (950k, A4, MicroChem Corp.) was added on top of the graphene. The Cu foil was etched away with FeCl$_3$ (CE-100, Transene) then the graphene/PMMA stack was rinsed in a series of deionized water baths until transfer to the prepared membranes. Once transferred, the PMMA was removed by soaking in 60 °C acetone for 1 h. After the graphene transfer, a 5 nm film of HfO$_2$ was grown via atomic layer deposition. To prepare the SiN$_x$ membranes for the next steps, the Si frame of the sample was glued to a carrier Si chip with PMMA (950k, A8, MicroChem Corp.). The prepared substrate was then coated with a negative tone hydrogen silsesquioxane resist (HSiQ, 6%, DisChem Inc.) at 100 nm. The sample was then exposed and patterned using the Elionix ELS G-100, an electron beam lithography tool. After exposure, the samples were developed in MF-321 for 90 s, with a 30 s rinse in DI water and then a 30 s rinse in IPA. The development process converts the exposed HSiQ to SiO$_x$. For metal deposition of the top, a metal mask was placed above the substrate to create electrically disconnected regions. The deposition consisted of a 2.5 nm chromium adhesion layer and 30 nm of gold. Following these processing steps, the graphene was found to be heavily hole-doped, similar to what has been observed in previous works[36,45]. Gate-dependent resistivity measurements showed an increase in resistance for positive gate bias, but no maximum resistance was observed that would indicate charge neutrality. These measurements also exhibited hysteresis, consistent with what has been observed elsewhere, and indicative of surface, interface, and substrate charge traps that can be populated with charge as $V_G$ is changed. At high biases, these traps can screen the applied gating field without doping the graphene, leading to deviations from the simple capacitance model that we use to estimate the graphene carrier density for a given $V_G$[57–59].

## Thermal Emission Measurements

The emission measurements were performed using a Bruker Vertex 70 FTIR, where thermal emission from a heated sample was used as the light source of the interferometer. The device was mounted on a home-built rotation stage, and thermal emission from the device is collected by the FTIR[44]. A carbon-nanotube source was used as our blackbody reference measurement. The finite size of the aperture creates a 3° acceptance angle, and there is also some uncertainty in the overall angle due to mechanical play in the stage holder and sample tilting within the sample holder. We estimate this uncertainly to be ≤3° based on measurements with an alignment laser reflected off of an unpatterned area of the sample surface.

## Optical Simulations

The frequency-dependent dielectric functions of $Al_2O_3$, Cr, Au and $SiO_x$ were taken from the Palik data[60]. The dielectric functions of $HfO_2$ and $SiN_x$ were obtained from infrared ellipsometry[30]. Heat-induced dielectric function change of $SiN_x$ is corrected through the higher-order F-P resonance peak which is insensitive to Fermi level modulation (see Supplementary Note 4). The graphene was modeled as a layer with zero thickness, and its optical conductivity was calculated by Kubo formula[61]. The carrier mobility of graphene is assumed to be 300 $cm^2$/Vs which is comparable to a previously reported value[30]. The reflection/transmission coefficients and absorption spectrum of the proposed structure were calculated by full-wave simulation with the finite element method. We determined the Fermi level of graphene at $V_G = 0$ V by comparing the calculated and measured frequency and angular emissivity spectra. This process aimed to minimize the difference in various parameters, including resonance frequency, intensity and full-width half maximum. Throughout the estimation process, we constrained the expected Fermi level to the range of -0.45 to -0.55 eV[45], consistent with the hole-doped state observed at the $V_G = 560$ V.

## Data availability

The experimental and theoretical data used to generate the figures in this manuscript and in the Supporting Information are available on Zenado [https://doi.org/10.5281/zenodo.10615359].

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

## Acknowledgements

J.S. and V.W.B. were supported by the Gordon and Betty Moore Foun-dation through a Moore Inventors Fellowship. M.F. was supported by Office of Naval Research award N00014-20-1-2356. M.A.K. and C.W. acknowledge support from the US Office of Naval Research, award N00014-20-1-2297. This work was also supported by the National Research Foundation of Korea (NRF) grants funded by the Ministry of Science and ICT (NRF-2022R1A2C2092095, S.K. and M.S.J.) and by the Ministry of Education (NRF-2022R1I1A1A01065727, S.K.).

## Author contributions

S.K., M.S.J. and V.W.B. conceived the idea. J.S. fabricated and char-acterized the devices, and performed the optical measurements. M.F. assisted in sample fabrication. M.A.K. and C.W. assisted in optical measurements. S.K. and M.S.J. conducted a theoretical analysis. S.K. conducted optical simulations and device optimization. L.S. and P.W. C.W. assisted in theoretical design and fabrication planning. J.S., S.K., M.S.J. and V.W.B. wrote the manuscript. M.S.J. and V.W.B. supervised the project.

## Competing interests

The authors declare no competing interests.
