## [Peer Review File · Nature Communications]

Electrostatic Steering of Thermal Emission with Active Metasurface Control of Delocalized ModesREVIEWER COMMENTS

Reviewer #1 (Remarks to the Author):

In this work, the authors propose and demonstrate a mechanism to electrostatically control the directional response of thermal emission. A Fabry-Perot resonance based delocalized mode was investigated for thermal beam steering by electrostatically tuning the effective surface admittance of the top metal-graphene meta-surface. The change in the Fermi level of graphene due to an external electrostatic field corresponded to a directional change of 16 degrees in the angular emissivity spectra at a fixed resonance frequency. This is an exciting result and the argument is well supported by theoretical backing as well. I am supportive of publication of this work but have a few comments that I hope are helpful.

1. Could the authors provide a full colormap emissivity spectra near the resonance frequency ($\omega = 1300 \text{ cm}^{-1} \sim 1700 \text{ cm}^{-1}$) as a function of angle ($0^\circ \sim 90^\circ$)? This may be helpful to better understand the full spectral/angular response of the metasurface architecture. It would also allow us to better understand how the whole mode shifts (both spectrally and directionally) when a gate voltage is applied.
2. The metal-graphene meta-surface that is delineated in the manuscript seems to include a MIM (metal-insulator-metal) geometry (between the top and bottom gold contacts) within it, could the authors provide an explanation of the possibility of localized MIM modes being excited through the geometry? It is not clear if possibly localized modes are affecting the angular beam steering of the device.
3. The authors argue that the small angular deviation they see in their experiments compared to calculations/ simulations is due to metasurface imperfections and averaging over the whole sample area. But isn't the fitting parameter a potential culprit here? For instance, there is already a noticeable deviation in peak angle for the 0V (experimental) and the -0.48 eV (calculation). Similarly there is a suppressed spectral peak shifting in the experimental relative to calculation as well. To me this suggests a systematic issue, perhaps attributable to voltage drops over other components of the device such that the actual gate voltage seen by the graphene layer is smaller than expected.
4. More generally, a brief justification on the validity of using the value of EF at $V_G = -0.560\text{V}$ as a fitting parameter would be useful. What does the theoretical model tell us when the initial value of EF was chosen for a different gate voltage?
5. Is there a simple dispersion relation-based model that shows the change of the angular response of the delocalized modes as a function of applied voltage or Fermi level of graphene? It is not necessary for the paper, but if a simple picture here is possible it might be an attractive means of understanding the beam position changing.
6. Finally, could the authors elaborate on the purpose/function of filling the gaps between the gold strips with SiOx?

Reviewer #2 (Remarks to the Author):

Controlling the direction of thermal emission is challenging while important for remote

sensing, space applications, and thermal management. This study used a graphene layer on top of a SiN_x dielectric whose back surface is coated with a 100 nm gold film to form an asymmetric Fabry-Perot structure. Patterned 30 nm Au film were deposited on the graphene layer to form a partially transparent mirror. Furthermore, the graphene layer can be electrostatically biased to change its Fermi level, thereby affecting the carrier concentration and conductivity. This will modify the resonance conditions (phase angle), causing the resonance emission direction to vary. A detailed theoretical model is provided and the fabrication processes are explained clearly. The experiments are very challenging but the results are convincing and supported by theoretical analysis. Nevertheless, the experimental results do not show as significant an angular variation (Fig. 3c) as that of the theoretical predictions (Fig. 3f). Yet, this is an original work and some further improvement may be achieved in the future as discussed in the manuscript. Overall, the manuscript is carefully prepared with high-quality figures and only a few typos.

Some comments are given below:

(1) In Methods A Fabrication. It may be better to start with a word rather than a number "2 um". Also, the sentence 2 um thick 5 mm x 5 mm SiN_x membranes on a 200 um thick Si frame were ... However, it appears that there are at least four layers: 2 um SiN_x, 100 nm Au, 2.5 nm Cr, 200 um Si. Another possibility is that the 200 um Si is etched first, like Fig. 1a, then Cr and Au are deposited on SiN_x. Perhaps the first couple of sentences can be modified. In Fig. 4a, the substrate is not shown. One should mention that the 100 nm Au makes the lower layer essentially opaque.

(2) Since the top layer are made of pattered gold strips (not stips as in the manuscript), I wonder whether this will cause any anisotropic effect like gratings. According to the effective medium theory, patterned metallic strips behave very differently for different polarizations. Perhaps the filling ratio (740/770) is so large that the anisotropic effect is very small.

Reviewer #3 (Remarks to the Author):

In this paper, Siegel et al. explored thermal emission through electrostatic gating and presented an intriguing research topic with potential for publication. While the study has merit, several concerns should be addressed for a comprehensive evaluation:

1.

The authors claim to have conducted electrostatic gating, but it is crucial to ensure proper execution by characterizing the leakage current as a function of gate voltage.

2.

The use of a high applied voltage (560 V) raises practicality concerns.

3.

Why did not the authors gate graphene via HfO₂, a high-K material?

4.

The authors simulated the value of the Fermi level. What do the values mean? Are they the energy difference between E_f and charge neutral point?

5.

Why the graphene is such heavily hole-doped? Can higher gate voltage, like 1000V, make the graphene electron-doped? How will the device perform if the graphene is electron-doped? Can the device be applied a voltage of 1000 or 1500 V? What is the theoretical breakdown voltage of the SiN? Will gating via HfO₂ help to tune the graphene to electron-doped?

6.

The role of Al₂O₃ in the study needs elucidation. What are the dielectric constants of Al₂O₃,

SiN, and the Al₂O₃/SiN bilayer?

7.

The schematic in the dashed frame in Fig 1b is unclear. What is the cyan oxide? Is that Al₂O₃? If so, where is SiN?

Response Letter

We appreciate the valuable comments of the Reviewers. We have carefully revised the manuscript in light of all the questions raised by the Reviewers. Our point-by-point responses to them are marked in blue.

Reviewer #1

In this work, the authors propose and demonstrate a mechanism to electrostatically control the directional response of thermal emission. A Fabry-Perot resonance based delocalized mode was investigated for thermal beam steering by electrostatically tuning the effective surface admittance of the top metal-graphene meta-surface. The change in the Fermi level of graphene due to an external electrostatic field corresponded to a directional change of 16° in the angular emissivity spectra at a fixed resonance frequency. This is an exciting result and the argument is well supported by theoretical backing as well. I am supportive of publication of this work but have a few comments that I hope are helpful.

We thank the reviewer for the thorough evaluation of our manuscript and for their useful questions and comments. Inspired by some of the reviewer's insightful suggestion, we have performed new, comprehensive calculations that explore how background doping effect overall angular tuning, and we have employed more rigorous fitting procedures to estimate the Fermi level position in our devices. The results of those calculations are discussed in our response to comments #3 and #4 below, and also discussed in the main text and methods section of our manuscript. All relevant theory figures have also been updated accordingly.

1. Could the authors provide a full colormap emissivity spectra near the resonance frequency ($\omega = 1300 \text{ cm}^{-1} \sim 1700 \text{ cm}^{-1}$) as a function of angle ($0^\circ \sim 90^\circ$)? This may be helpful to better understand the full spectral/angular response of the metasurface architecture. It would also allow us to better understand how the whole mode shifts (both spectrally and directionally) when a gate voltage is applied.

We agree with the reviewer that a full angle- and frequency-dependent colormap of emissivity would be useful in illustrating how our device works. We have performed new simulations at multiple gate voltages to generate the colormaps at Fermi levels of -0.38 and -0.68 eV as shown in the Fig. R1. We have also included below the experimental colormaps generated from emissivity measurements taken from 0 to 30° in 2° steps. We note that Fig. 3(c, f) of the main text show both experimental and theoretical data that are subsets of these plots and indicate how emissivity changes for the largest changes in gate-voltage (i.e. Fermi level) and experimentally-achievable measurement angle.

The theoretical and experimental emissivity colormaps are now included as Fig. S21 in Supplementary Information Note 7.

Figure R1. (a, b) Theoretical and (c, d) experimental colormaps of emissivity as a function of frequency and angle for graphene Fermi levels of -0.38 eV [$V_G = 560 \text{ V}$] (left) and $V_G = -0.68 \text{ eV}$ [$V_G = -560 \text{ V}$] (right)

2. The metal-graphene meta-surface that is delineated in the manuscript seems to include a MIM (metal-insulator-metal) geometry (between the top and bottom gold contacts) within it, could the authors provide an explanation of the possibility of localized MIM modes being excited through the geometry? It is not clear if possibly localized modes are affecting the angular beam steering of the device.

The reviewer is correct that our structure can support MIM plasmonic modes, however, the wavelength of those modes (for the target frequency range in this work) is much larger than the metal bar widths as well as the periodicity of the structures investigated in this work. That mismatch prevents resonant absorption into the MIM modes for the device dimensions used in this manuscript. In order to provide support for this supposition, we have performed full-wave, finite element simulations over a spectral range of $1300 - 1700 \text{ cm}^{-1}$ with bar widths varying from $0.8 - 1.2 \mu\text{m}$, similar to those used in the main manuscript, and from $3.2 - 3.6 \mu\text{m}$, which is wide enough to potentially support MIM resonances in the target spectral range. The results of those simulations are shown in the Fig. R2. As seen in those simulations, when the bar width is narrow $0.8 - 1.2 \mu\text{m}$ only a single Fabry-Perot (FP) resonance is observable, while for the larger widths, both FP and MIM resonances are apparent.

The FP and MIM resonances can be distinguished in two different ways in these simulations. First, the FP resonance frequency is not strongly dependent on the bar width, since its wavelength is largely dictated by the dielectric thickness. The small energy shifts that are observed in the FP resonance are due to the effect of scattered phase on the graphene-metal metasurface, which changes slightly with bar width. The MIM modes, meanwhile, are observed to be strongly dependent on the bar width since the MIM resonant wavelength is directly related to the cavity length dictated by the width of the metal bars. Second, our simulations allow us to directly plot the field profiles of the FP resonances and MIM resonances, which are included as insets in the figure below. Those field profiles clearly show that the weakly dispersive mode is

a vertical standing wave in the dielectric layer, consistent with a FP mode, while the highly dispersive mode is a lateral standing wave, consistent with a MIM resonance.

In summary, while our structure does support MIM modes, the dimensions we use in our metasurface forces any MIM resonances to occur at frequencies much higher than the FP resonances that we focus on, and the ultimate effect of the MIM resonances is minimal.

Figure R2. Normal angle emissivity of thermal steering device described in the main text, with the metal bar width varying from (a) 0.8 to 1.2 μm and (b) 3.2 to 3.6 μm . Red lines indicate the emissivity peaks associated FP modes are drawn in red, while those associated with MIM modes are blue. Only FP modes occur between 1300 and 1700 cm^{-1} for 0.8 to 1.2 μm bar widths.

3. The authors argue that the small angular deviation they see in their experiments compared to calculations/simulations is due to metasurface imperfections and averaging over the whole sample area. But isn't the fitting parameter a potential culprit here? For instance, there is already a noticeable deviation in peak angle for the 0 V (experimental) and the -0.48 eV (calculation). Similarly, there is a suppressed spectral peak shifting in the experimental relative to calculation as well. To me this suggests a systematic issue, perhaps attributable to voltage drops over other components of the device such that the actual gate voltage seen by the graphene layer is smaller than expected.

We thank the reviewer for the opportunity to clarify this important issue. Below we discuss different potential sources of systematic errors in fitting the carrier density vs. applied gate voltage.

Possibility of significant voltage drop across other device components: It is certainly a valid point that, if the voltage drops over other components of the device is significant, it could result in inefficient gating and thus diminish the angle steering range and the associated spectral peak shift. However, we believe this scenario is unlikely because the measured resistance across the gate dielectric ($> 10^2 \text{ M}\Omega$) is many orders of magnitude higher than any electrical contact in our circuit and the resistance of the graphene sheet (measured to be $\sim 200 \Omega$). We observe that the combined $\text{SiN}_x/\text{Al}_2\text{O}_3$ dielectric stack exhibits leakage that increases with applied gate voltage and higher temperatures, however, the resistance across the gate remained greater than 200 $\text{M}\Omega$ in all measurements (see Fig. R3) Thus, the gate voltage drop still occurs almost entirely across the $\text{SiN}_x/\text{Al}_2\text{O}_3$, despite the small leakage current.

Figure R3. Measurement of leakage current across the SiN_x/Al₂O₃ layers between the graphene sheet and the bottom Au backgate. The device temperature is maintained at 250°C during measurement.

Dependence of gate capacitance on temperature and gate voltage: Gating could also become inefficient when the gate capacitance drops with increasing temperature or gate voltage. The capacitance between the graphene and the backgate (i.e. gold reflector) is determined by the static dielectric constants of the dielectric materials. For the SiN_x substrates used in this work, previous measurements have shown that the dielectric constant increases only slightly with temperature up to 250°C, and is unchanged by the applied gate voltage (see Supplementary Figure 2 in in Ref. [*Nat. Commun.* 6, 7032 (2015)]).

These measurements provide justification for the basic capacitance model we use in estimating how carrier density changes with applied gate voltage, which assumes the dielectric constants of the SiN_x/Al₂O₃ to be independent of temperature and gate voltage.

Possible causes of reduced carrier density modulation: We note that there is a possibility that the static dielectric constant of low stress LPCVD SiN_x is lower than our expectation. In our current calculation, we assume its dielectric constant as 7.5, as derived in our previous work [*Phys. Rev. B* 90, 165409 (2014)]. However, since the dielectric constant of the SiN_x membrane produced by Norcada has a variance of 1 [*X-Ray Microscopy Windows Specification Sheet*, Norcada, www.norcada.com], the gate capacitance of our device possesses ~15% uncertainty.

Beyond the simple electrostatic model that we use, however, are charge traps and atmospheric impurities on or in the SiN_x/Al₂O₃ which are known to change their charge state depending on the applied gate voltage. The effects of such impurities are discussed in our main text and in Methods section A - they act to effectively screen the applied gating field without changing the graphene carrier density, thus leading to deviations from the simple capacitor model. Our evidence for such impurity states is the hysteresis that we observe in resistance vs. gate voltage measurements, which is consistent with previous studies that systematically investigated such charge traps and impurities.

We have added this discussion of these issues to the Supplementary Information Note 5, and have also discussed them in more detail in the main text.

4. More generally, a brief justification on the validity of using the value of E_F at $V_G = -560$ V as a fitting parameter would be useful. What does the theoretical model tell us when the initial value of E_F was chosen for a different gate voltage?

We thank the reviewer for their insightful question. Recognizing that the carrier density modulation via electrostatic gating has a certain degree of uncertainty as stated in the response to the comment #3, we realized that it would be more reasonable to use the middle point (E_F at $V_G = 0$), instead of an end point (E_F at $V_G = -560$ V), as the reference point for our fitting. By using E_F at $V_G = 0$ as our new fitting parameter, the E_F at $V_G = -560$, 0, and +560 V, are now estimated as -0.68 , -0.55 , and -0.38 eV, slightly different than our previous estimates of -0.62 , -0.48 , and -0.27 eV, respectively. The fitting procedure is illustrated in the figure below, where we calculate the frequency-dependent absorption of the structure for different Fermi levels and compare to the experimental spectra obtained at $V_G = 0$.

Figure R4 The absorption spectra corresponding to various reference Fermi levels and the measured emissivity spectrum (Exp) at $V_G = 0$.

We first note that the new (as well as the old) estimated E_F values are consistent with other experimental observations. The graphene used in these experiments were prepared using the same procedure as our previous work on graphene nanostructures. [*ACS Photonics* 8, 1277 (2021)]. In that work, we found that the graphene transfer process doped our graphene to an initial Fermi level (i.e. $V_G = 0$ V) of $-0.45 \sim -0.55$ eV due to the FeCl etchant used to dissolve the copper foil the graphene was grown on, which is consistent with our estimate ($E_F = -0.55$ eV at $V_G = 0$ V). Furthermore, in resistance vs. gate voltage measurements taken from our device, shown Fig. R5, the resistance continually increases as the gate voltage increases, without reaching a maximum that would indicate that the graphene has been doped to charge neutrality (such as the resistance maxima were observed in [*Nat. Commun.* 6, 7032 (2015)] for differently prepared samples, as well as in many other publications). This result also agrees with our estimate that $E_F < 0$ eV at $V_G = 560$ V.

Figure R5. Resistance vs. gate voltage measurements of the graphene in the thermal steering device obtained via a single gate voltage sweeps over a small voltage range (left) and an average of many voltage sweeps over an extended range (right)

Ultimately, however, the estimated changes in initial Fermi level make little difference in the calculated angular tuning or frequency-dependent emissivity of the device. To illustrate this, the following figure compares the experimental frequency- and angular-spectra to the calculated ones with the original ($E_F = -0.62, -0.48, \text{ and } -0.27 \text{ eV}$) and new ($E_F = -0.68, -0.55, \text{ and } -0.38 \text{ eV}$) Fermi level estimates. The new E_F estimates do result in a slightly better match between theory and experiment in terms of expected angular tuning as well as modulation depth. But, there is still a noticeable mismatch that we attribute to a combination of various causes stated in the main text and described in our response to the comment #3. In the revised manuscript, we have updated Figures 2 and 3 in the main text by using the new E_F estimates.

Figure R6. Comparison between calculated for two different reference Fermi levels and measured emission spectrum for various conditions. (a, d, g) Frequency emission as a function of gate voltages for normal direction and (b, e, h) oblique angles at a constant applied gate voltage -560 V. (c, f, i) The angular emission spectra at the (c) 1508, (f) 1498, and (i) 1503 cm^{-1} .

One interesting outcome from these calculations is that they indicate that one simple method to achieve improved angle tunability and modulation depth would be to use graphene that has less initial doping as shown in Fig. R6 and R7. For example, if $E_F = -0.4$ eV for $V_G = 0$ V, which corresponds to $E_F \sim 0$ eV for $V_G = +560$ V, we expect to achieve 6° larger angular steering range compared to the case of -0.55 eV initial doping. We have now included a mention of this in our manuscript, and we thank the reviewer for their helpful question that allowed us to realize this new point regarding the performance of our device.

Figure R7. Calculated angle steering range as a function of the initial doping level of graphene.

5. Is there a simple dispersion relation-based model that shows the change of the angular response of the delocalized modes as a function of applied voltage or Fermi level of graphene? It is not necessary for the paper, but if a simple picture here is possible it might be an attractive means of understanding the beam position changing.

As the Reviewer pointed out, a simple and intuitive model would definitely help understanding the relation between the Fermi level and the peak emission angle (or equivalently, peak absorption angle). In the manuscript, we describe the beam steering behavior as a Fano interference between the non-resonant direct reflection from the top surface (r_{direct}) and the resonant Fabry-Perot reflection formed by multiple reflections inside the dielectric spacer (r_{FP}). Since r_{direct} does not depend much on the Fermi level and the angle as shown in Fig. 2(c) and Fig. S7(d), the condition for the destructive interference (i.e., the condition for the peak emission) is mainly determined by r_{FP} , which can be expressed analytically as follows (see also Supplementary Note 3):

$$r_{\text{FP}} = \frac{t_{\text{in}} t_{\text{out}} r_{\text{bottom}} e^{2ik_{\text{out}}h}}{1 - r_{\text{top}} r_{\text{bottom}} e^{2ik_{\text{out}}h}} \approx - \frac{t_{\text{in}} t_{\text{out}} e^{2ik_{\text{out}}h}}{1 + r_{\text{top}} e^{2ik_{\text{out}}h}}$$

Here, since noble metals behave like a perfect electric conductor at mid-IR, the reflection coefficient at the bottom Au surface, r_{bottom} , can be approximated to -1. As the angle θ increases, the propagation phase

across the dielectric layer, $k_{\text{out}}h = kh \cos \theta$, decreases. To compensate for this, E_F should be decreased to make the metasurface more capacitive and achieve a higher transmission phase $\arg \{t\}$.

We note that the reflection and transmission coefficients are dependent on the surface admittance of the graphene based metasurface \tilde{Y}_s as follows (see also Supplementary Note 1):

$$r_{12} = \frac{n_1 \cos \theta_2 - n_2 \cos \theta_1 - \tilde{Y}_s \cos \theta_1 \cos \theta_2}{n_1 \cos \theta_2 + n_2 \cos \theta_1 + \tilde{Y}_s \cos \theta_1 \cos \theta_2}$$

$$t_{12} = \frac{2n_1 \cos \theta_1}{n_1 \cos \theta_2 + n_2 \cos \theta_1 + \tilde{Y}_s \cos \theta_1 \cos \theta_2}$$

The surface admittance \tilde{Y}_s is a function of the graphene conductivity σ_G , which is determined by the Fermi energy of graphene, E_F , and can be roughly approximated as [ACS Nano 14, 1166 (2020), Junhyung Kim, MS thesis, Korea Advanced Institute of Science and Technology (2020)]

$$\tilde{Y}_s = \frac{1}{Y_0} \left[\frac{-i\omega(L_m + L_k)}{P} + \frac{1}{-i\omega P(C_g + C_c) + (P/g)\sigma_G} \right]^{-1}$$

where Y_0 is the free space admittance, P and g are the period and the gap width, respectively. The magnetic (L_m) and kinetic (L_k) inductance and the gap (C_g) and coplanar (C_c) capacitances can be expressed by using structural parameters and material permittivities as discussed in [ACS Nano 14, 1166 (2020) and Junhyung Kim, MS thesis, Korea Advanced Institute of Science and Technology (2020)], although our device structure is more complicated due to the Au/SiO_x bilayer filling the gap. Since the dependence of σ_G on E_F can also be analytically written using the random phase approximation [J. Phys. Conf. Ser. 129, 012004 (2008)], in principle, it is possible to come up with a semi-analytic model by weaving all these components together. However, we find that this full model is not comprehensive or accurate enough to be included in the main text. We also note that, since we describe the relation between the emission angle and the Fermi level at a fixed frequency, an energy-momentum dispersion relation is not much relevant.

The mechanism we describe above is distinct from previous static directional thermal emission experiments that utilized grating structures to selectively outcouple long-lived surface modes [Nature 416, 61–64 (2002)]. For those experiments, a dispersion model could cleanly describe the directionality of out-coupled modes, but in our device the graphene/metal metasurface does not act as a grating to match free-space modes with surface modes. Instead, it acts as a boundary with gate-dependent reflection and transmission phases.

6. Finally, could the authors elaborate on the purpose/function of filling the gaps between the gold strips with SiO_x?

The filled gaps in our structure are a result of the particular fabrication process based on a negative-tone HSQ resist that we utilize to create the device. This device requires small, precise features (~30nm) to be written over large, mm-scale areas. Writing such patterns presents two challenges to electron beam lithography. First, the write time using a conventional PMMA resist over a mm-scale area can be very long (~1 day) and cost-prohibitive unless large beam currents are used. However, if large beam currents are used, small features cannot be cleanly written. Second, a conventional positive-tone resist would require

exposing the electron beam to all areas of the sample covered with metal, which represents most of the sample in our structure. Exposing such large areas would create dramatic proximity effects that would also prevent small structures from resolving. As such, we used a negative tone HSQ resist which develops with much smaller doses than PMMA, and would only require exposing the gap part of our structure, which covers less surface area. HSQ, when exposed and developed, leaves behind a SiO_x material on the surface, which is typically removed with hydrofluoric acid (HF). However, in our structure the use of HF would damage the graphene and possibly delaminate the gold structures, so we chose to leave the SiO_x in place.

Reviewer #2

Controlling the direction of thermal emission is challenging while important for remote sensing, space applications, and thermal management. This study used a graphene layer on top of a SiN_x dielectric whose back surface is coated with a 100 nm gold film to form an asymmetric Fabry-Perot structure. Patterned 30 nm Au film were deposited on the graphene layer to form a partially transparent mirror. Furthermore, the graphene layer can be electrostatically biased to change its Fermi level, thereby affecting the carrier concentration and conductivity. This will modify the resonance conditions (phase angle), causing the resonance emission direction to vary. A detailed theoretical model is provided and the fabrication processes are explained clearly. The experiments are very challenging but the results are convincing and supported by theoretical analysis. Nevertheless, the experimental results do not show as significant an angular variation (Fig. 3c) as that of the theoretical predictions (Fig. 3f). Yet, this is an original work and some further improvement may be achieved in the future as discussed in the manuscript. Overall, the manuscript is carefully prepared with high-quality figures and only a few typos.

Some comments are given below:

1. In Methods A Fabrication. It may be better to start with a word rather than a number "2 μm". Also, the sentence 2 μm thick 5 mm x 5 mm SiN_x membranes on a 200 μm thick Si frame were ... However, it appears that there are at least four layers: 2 μm SiN_x, 100 nm Au, 2.5 nm Cr, 200 μm Si. Another possibility is that the 200 μm Si is etched first, like Fig. 1a, then Cr and Au are deposited on SiN_x. Perhaps the first couple of sentences can be modified. In Fig. 4(a), the substrate is not shown. One should mention that the 100 nm Au makes the lower layer essentially opaque.

We thank the reviewer for the suggestion. We note that the 200 μm Si is only present on the edge of the chip, and acts as a supporting frame for the membrane. The membrane itself represents the active area of the device. To clarify the fabrication process we modified the following sentence

“2 μm thick, 5 mm x 5 mm SiN_x membranes on a 200 μm thick Si frame were purchased from Norcada. Metal deposition of the back-reflector consisted of a 2.5 nm chromium adhesion layer and 100 nm of gold.”

to

“SiN_x membranes (2 μm thick and 5 mm x 5 mm wide) on a 200 μm Si frame were purchased from Norcada. To the backside of the SiN_x membrane, we deposited a 2.5 nm chromium adhesion layer followed by a 100 nm of gold, which makes the lower layer opaque and reflective.”

Figure 4(a) illustrates the structure of the active area and thus the Si frame was not included. To make this point clear, we have modified the caption of Fig. 4 as follows:

“Fig. 4. (a) Schematic of the active area structure to obtain potentially achievable maximum emission angle steering performance. The bilayer Au/SiO_x is eliminated and geometry parameters are modified. (b) The angular emission spectra of the optimized device for Fermi level of 0 and 0.6 eV at $\omega = 1614\text{cm}^{-1}$ ”

2. Since the top layer are made of patterned gold strips (not strips as in the manuscript), I wonder whether this will cause any anisotropic effect like gratings. According to the effective medium theory, patterned metallic strips behave very differently for different polarizations. Perhaps the filling ratio (740/770) is so large that the anisotropic effect is very small.

The reviewer is correct to identify that the structure is polarization dependent and our device only functions for TM polarization. In the TE polarization, the structure acts as a mirror with negligible emission. Our measurements made use of a polarizer. In the Fig. R8, we can observe the change in the raw signal as a function of polarizer angle *with respect to the long axis of the gold bars*. When aligning the polarizer parallel to the long axis of the metal slit ($\theta \rightarrow 0^\circ$, TE polarization), the emission signal asymptotically converges to 0. Conversely, the emission signal gradually increases as the polarizer aligns parallel to the short axis of the metal slit ($\theta \rightarrow 90^\circ$, TM polarization).

Figure R8. Polarization-dependent emissivity measurements of the device, obtained at 250°C.

Reviewer #3

In this paper, Siegel et al. explored thermal emission through electrostatic gating and presented an intriguing research topic with potential for publication. While the study has merit, several concerns should be addressed for a comprehensive evaluation:

1. The authors claim to have conducted electrostatic gating, but it is crucial to ensure proper execution by characterizing the leakage current as a function of gate voltage.

We thank the reviewer for the opportunity to clarify this issue. According to the reviewer's suggestion, we present the measured leakage current as a function of gate voltage as shown in the Fig. R3. The resistance of the dielectric spacer is typically on the order of 10^2 M Ω , which is many orders of magnitude higher than that of the graphene layer (100~300 Ω). Consequently, we can affirm that the voltage drop of the gate occurs across the dielectric, and that the electrostatic gating is sufficiently close to the ideal situation with negligible leakage current. Detailed, gate-dependent gate leakage measurements of our device are now included in section S5 of the supplement to clarify this issue.

2. The use of a high applied voltage (560 V) raises practicality concerns.

We agree with the reviewer that the use of a high gate voltage makes the device less practical. We note that, however, the main purpose of this paper is to demonstrate the possibility of dynamic thermal steering with electrostatic gating, and the actual operational voltage can be significantly reduced by modifying the gating scheme. For example, as shown in [*Commun. Mater.* 1, 7 (2020), *Light Sci. Appl.* 7, 51 (2018)], by inserting a transparent conducting layer near the top graphene membrane, the gate voltage required to achieve the same level of Fermi energy can be reduced by orders of magnitudes with a marginal perturbation of the device's optical characteristics.

To emphasize the possibility of reducing the gate voltage, we have added the following sentences right before the Conclusion section.

“We note that the required gate voltage for device operation can be significantly reduced by modifying the gating scheme. For example, as demonstrated in Ref. 54 and 55, by inserting a transparent conducting layer near the top graphene membrane, the gate voltage required to achieve the same level of Fermi energy can be reduced by orders of magnitudes with a marginal perturbation of the device's optical characteristics”

3. Why did not the authors gate graphene via HfO₂, a high-K material?

As the reviewer pointed out, there exists a possibility to achieve larger Fermi energy modulation by employing a high-K gate dielectric such as HfO₂, and that is certainly an interesting direction to pursue in the future studies. In our work, we chose to adopt SiN_x membranes because it is a high-quality, large-area, few-micron-thick gate dielectric that is commercially available. One can grow a high-quality HfO₂ film in

large-area through atomic layer deposition, but the process is slow to achieve a few-micron-thick film. It is also possible to employ a transparent gate electrode mentioned above and add a few tens of nanometers of HfO₂ as a gate dielectric, which would be an interesting future research.

4. The authors simulated the value of the Fermi level. What do the values mean? Are they the energy difference between E_F and charge neutral point?

The reviewer is correct that in our manuscript ‘Fermi level’ refers to the energy difference between E_F and the charge neutral point. The Fermi energy, E_F , and the carrier concentration, n , are related as $n = E_F^2 / \pi \hbar^2 v_F^2$, where $v_F \sim 10^6$ m/s is the Fermi velocity of graphene.

5. Why the graphene is such heavily hole-doped? Can higher gate voltage, like 1000V, make the graphene electron-doped? How will the device perform if the graphene is electron-doped? Can the device be applied a voltage of 1000 or 1500 V? What is the theoretical breakdown voltage of the SiN? Will gating via HfO₂ help to tune the graphene to electron-doped?

The origin of hole-doping: Graphene is heavily hole doped during the Cu etching process using FeCl₃ [*ACS Photonics* 8, 5, 1277 (2021)]. Alternative etching and cleaning methods can be used to reduce the hole-doping, such as using ammonium persulfate as the Cu-etchant [*Nat. Nanotechnol.* 5, 574 (2010)], but in these devices the large hole doping works to our advantage as it allows for larger changes in the magnitude of carrier density. A graphene sheet with no doping, for example, would reach equally large values of electron and hole doping for positive and negative gate voltages, respectively, rather than fluctuating between a small and very large hole doping, as we achieve here. In our reply to Reviewer 1 (comment #4) we note that there is an optimal background doping level that enables maximum tunability for a given range of applied gate voltages, and we have added a discussion of that finding in our manuscript and in the supplementary section.

Dielectric strength of SiN: For low-stress silicon nitride, the dielectric strength ranges from 100 to 500 V/ μ m, making a 2 μ m thickness film potentially suitable for operation at 1000 V. However, the actual breakdown voltage of the film varies sample to sample. In our experiment, our methods of contacting the sample (i.e., wirebonding) weakened the dielectric strength of SiN_x and thus restricted the range of gate voltage up to 560 V.

Possibility to achieving n-doped graphene: A gate voltage of 1000 V is enough to make the graphene electron-doped (i.e., positive Fermi level). Gating via HfO₂ can help to tune the graphene to electron-doped regime due to its high dielectric constant. However, the optical response of the device is mainly determined by the absolute carrier density, not by the carrier type. Therefore, the advantage of achieving an electron-doped regime would be marginal.

6. The role of Al₂O₃ in the study needs elucidation. What are the dielectric constants of Al₂O₃, SiN, and the Al₂O₃/SiN bilayer?

The Al₂O₃ layer primarily enhances the stability of electrostatic gating. The low-stress silicon nitride layer inherently includes pinholes, which increase leakage current and lower the breakdown voltage. Adding a thin layer of Al₂O₃ blocks pinholes and leads to a more stable gate dielectric. The static dielectric constants of Al₂O₃ and SiN_x are 9.3 and 7.5, respectively [*Adv. Mater. Interfaces* 6, 1802055 (2019), *Phys. Rev. B* 90, 165409 (2014)], but the capacitance change caused by the presence of Al₂O₃ is negligible since it is two orders of magnitude thinner than SiN_x layer. In terms of optical properties, the Al₂O₃ layer modifies the propagation phase accumulation of Fabry-Perot modes, but this effect is negligible due to its very small optical path length.

7. The schematic in the dashed frame in Fig 1b is unclear. What is the cyan oxide? Is that Al₂O₃? If so, where is SiN?

As the reviewer pointed out, the labels in the original Fig. 1b was ambiguous. The cyan layer in the schematic represents a group of dielectric layers composed of SiN_x (2 μm) and Al₂O₃ (30 nm). To make this point clear, we label it as 'dielectric spacer' in the revised figure.

Figure R9. Revised Figure 1. The cyan layer is now labeled as ‘Dielectric spacer’.

REVIEWERS' COMMENTS

Reviewer #1 (Remarks to the Author):

The authors have done an excellent job responding to reviewer comments and I believe the manuscript is ready for publication at this time.

Reviewer #2 (Remarks to the Author):

I do not have any further comments. I think it can be published as is.

Reviewer #3 (Remarks to the Author):

I think the authors addressed most of issues raised by me, and I am ok to suggest a publication. Just the extremely large leakage current is a concern. In order to properly present the results to the readers, maybe the authors shall at least plot to currents in log, following the typical protocol. The results shall be included either in the SI or main text.